# ZipZap: Efficient Training of Language Models for Ethereum Fraud Detection

## ABSTRACT

Language models (LMs) have demonstrated superior performance in detecting fraudulent activities on Ethereum. Nonetheless, the sheer volume of Ethereum data results in excessive memory and computational costs when training LMs from scratch, limiting their capabilities to scale to a large magnitude for practical applications. In this paper, we present ZipZap, a framework tailored to achieve both parameter and computational efficiency when training LMs on Ethereum-centric data. First, with the *frequency-aware* compression, ZipZap is able to compress an LM down to a mere 6% of its initial size with an imperceptible performance dip. This technique correlates the embedding dimension of an address with its occurrence frequency in the dataset, motivated by the observation that embeddings of low-frequency addresses are insufficiently trained and thus negating the need for a uniformly large dimension for knowledge representation. Second, ZipZap accelerates the speed through the *asymmetric* training paradigm: It performs transaction dropping and cross-layer parameter-sharing to expedite the pre-training process, while revert to the standard training paradigm for fine-tuning to strike a balance between efficiency and efficacy, motivated by the observation that the optimization goals of pre-training and fine-tuning are inconsistent. In addition, extensive evaluations on real-world, large-scale datasets demonstrate that ZipZap delivers notable parameter and computational efficiency improvements for LMs tailored for Ethereum data. Our implementation is available at: https://github.com/Anonymous0925/ZipZap.

## 1 INTRODUCTION

Blockchain has given rise to a wide range of fraudulent activities [6, 14, 15, 36]. Take phishing scams [36], one of the most prevalent frauds [30] on Ethereum for example: victims often receive deceptive messages through email or social media that entice them to click on deceptive links, which authorize transactions to transfer Ether or tokens to fraudster's accounts. Another fraud is money laundering using zkSNARK [11]-powered coin-mixing services [24, 35] like Tornado Cash, which mitigates the traceability of transactions made between two accounts owned by the same fraudsters. The key to detecting such frauds lies in representing and analyzing the behavioral patterns of fraudulent accounts, either to differentiate them from legitimate ones or to identify similar accounts that are both controlled by the same fraudsters.

Many previous studies [15, 21, 29, 36] have shown that representing accounts in a latent space based on their transaction relationships using representation learning techniques [18, 23, 32], and detecting fraud in the latent space, is a promising and generalized solution. Recently, language models, renowned for their superior sequential modeling ability, have established a new state-of-the-art [15] over previous graph-based methods [21, 29, 36]. Although these approaches reach good performance on small datasets, they

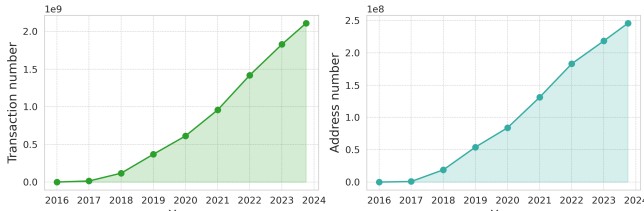

**Figure 1: Cumulative numbers of transactions and addresses on Ethereum across years.**

fall short in large-scale applications due the immense cost on memory and computation. As illustrated in Figure 1, there are approximately 2.1 billion transactions and 250 million addresses recorded on Ethereum as of October 2023 [8], and the numbers keep increasing over time. Assigning a 128-dimension embedding vector to each Ethereum address to represent its information would lead to 32 billion parameters in total, which would be prohibitively expensive for the majority of researchers, let alone the computational cost of training models on vast amount of transaction data from scratch.

**Scope and Contributions:** In this paper, we propose a framework, coined as ZipZap, designed to enhance parameter efficiency in language models and computational efficiency during their training. In our scenario, a language model (LM) serves as a sequence encoder that extracts account representations from sequences, which are constructed from accounts' historical transactions. LMs are initially pre-trained to capture co-occurrence relationship of transactions like BERT [7], GPT [25] and ELECTRA [5], and subsequently fine-tuned with a cascaded MLP classifier for downstream tasks.

To realize parameter-efficiency, we first identify that the bottleneck of parameter lies in the address embedding lookup table, whose parameter number scales linearly to the number of addresses, constituting 99% of the entire LM when the number of addresses approaches the million-level. Fortunately, a characteristic that can be exploited for streamlining is that the frequency of addresses follows a power-law distribution [20], indicating that the majority of parameters in the address embedding table are trained at a very low frequency, given that an address embedding can only be trained when its associated address appears at the transactions. This observation implies that it is unnecessary to allocate a uniformly large dimension to every address embedding. Instead, we propose the *frequency-aware compression* technique that positively correlates the dimension of the address embeddings with the occurrence frequency of their associated addresses through address space partitioning and dimension decay. This approach yields a remarkable compression rate (6%) with a negligible performance downgrade and accelerates training due to the reduction in backward gradient calculations.

To further expedite the training, another characteristic that can be harnessed is the inconsistencies between optimization goals and computation costs of pre-training and fine-tuning. We design

acceleration strategies specifically for pre-training to achieve computational efficiency as it accounts the majority time of training, while revert to the standard training paradigm during fine-tuning to preserve the effectiveness in downstream tasks. For example, we propose transaction dropping strategy for the pre-training stage, motivated by the observation that dropping repetitive transactions has no effect on transaction co-occurrence captured by the pre-training task, yet offers considerable computational conservation since the length of sequence exhibits a quadratic relationship to the time complexity of Transformer computation. Conversely, fine-tuning is conducted on recovered transaction sequences to fully capture the temporal patterns inherent in transactions, prioritizing effectiveness over efficiency. This strategy, named *asymmetric training*, allows ZipZap to enjoy efficiency during training without compromising effectiveness in downstream tasks.

Comprehensive experiments demonstrate that ZipZap represents a remarkable advancement over the state-of-the-art in both terms of parameter and computational efficiency: ZipZap streamline the original LM down to mere 6% with a marginal performance loss, and delivers up to *3* times speed during pre-training on large-scale real-world datasets. Moreover, a 1.44 absolute percentage gain in the $F_1$ metric is observed, which is on par with the original LM in detecting phishing scams, one of the most pervasive frauds on Ethereum [30].

In summary, this paper makes three original contributions:

- We present ZipZap, a framework that offers both parameter efficiency and computational efficiency for training LMs to facilitate practical Ethereum fraud detection.
- ZipZap enhances parameter efficiency in LMs by frequency-aware compression, which significantly reduces 94% of parameters of the original LM with an imperceptible performance dip.
- ZipZap strikes a well balance between efficiency of pre-training and efficacy on the downstream tasks via the asymmetric training paradigm. Along with reduced parameter, ZipZap offers up to **3** times speed up on large-scale datasets.

## 2 BACKGROUND AND RELATED WORK

### 2.1 Efficient Transformer Pre-training

To realize computational efficiency, the most straightforward way is to employ efficient Transformers, such as Performer [4], Linformer [34], Longformer [2], and Big Bird [39]. Another method is Progressive Stacking [10, 37], which takes advantage of the high similarity of cross-layer attention distribution to progressively stack shallow BERT models to generate deeper ones. Token dropping techniques [13, 26] can also improve computational efficiency by discarding or bypassing unimportant tokens, however, determining which tokens to drop without hurting performance can be challenging. Some learning-based methods[13, 17, 38] inevitably introduce extra computation, making them less efficient for training, or only suitable for inference.

To achieve parameter-efficiency, various techniques have been proposed. For example, ALBERT [19] factorizes the embedding layer and shares parameters across layers, resulting in a reduction in memory consumption. GroupReduce [3] partitions the language vocabulary into disjoint blocks and applies weighted SVD to achieve a low-rank approximation. Additionally, several works [9, 16, 22, 41, 42] from the recommendation field utilize neural architecture search (NAS) and reinforcement learning (RL) to learn variable embedding sizes. Among these, Learnable Embedding [22] shows the best performance with learnable soft-threshold pruning technique. Nevertheless, learning-based methods require the initiation of a large model at the start of training and entail considerable additional computation to determine the optimal configuration.

### 2.2 Terminology

*Externally owned account (EOA):* An EOA refers to an Ethereum account that is controlled by a user who has access to the account's private key. An EOA represents an individual user, and only EOAs can initiate transactions.

*Contract account:* A contract account represents a smart contract program deployed on Ethereum, which can be triggered by transactions issued by EOAs to achieve functionalities pre-defined in its code. Both EOAs and contract accounts are identified by an address, which is a 42-character hexadecimal string.

*Transaction:* Transactions are cryptographically signed data messages that contain a set of instructions, which can be interpreted to sending Ether between accounts or triggering a smart contract. A transaction consists of several elements:

- *Sender*: Address of the EOA that initiates the transaction.
- *Recipient*: Address of the account that receives the transaction.
- *Amount*: Amount of Ether being sent or received in the transaction.
- *Data*: Data used to specify the function of a smart contract to be called or the arguments to be passed.
- *Timestamp*: Timestamp of when the transaction was logged on the blockchain.

## 3 TRAINING LANGUAGE MODELS

To provide some backgrounds, we introduce a standard paradigm of pre-training a BERT-like LM [7, 15] on Ethereum data from scratch, and fine-tuning it for downstream fraud detection tasks.

### 3.1 Sequence Construction

As illustrated in Figure 2, an EOA has its own transaction sequence, which is constructed from the transactions the account has involved either as the sender or the recipient, with transactions sorted by timestamp. A *dummy* self-transaction is placed at the head of the sequence, its address feature set to the EOA's own address. This serves the dual purposes of incorporating self-address information into the model and facilitating the gathering of global information during self-attention computation. Each transaction has several features such as address, timestamp, position, amount, *etc*.

### 3.2 Model Architecture

*3.2.1 **Embedding Layer:*** Transaction features are encoded into embedding vectors via embedding lookup tables. As illustrated in Figure 3, we convert a 42-character hexadecimal address into an index *i* using a string-to-integer mapping, then retrieve the *i*-th embedding from the address embedding lookup table, which is a *d*-dimension address embedding vector that represents the address. Each type of features has its own embedding lookup table with $V \cdot d$

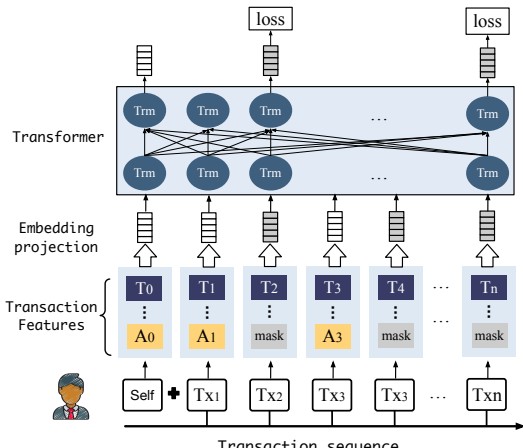

**Figure 2: Model architecture and pre-training task of a BERT-like LM.**

parameters, where $V$ is the total number of possible discrete values for that specific feature. For the address feature, $V$ can be in the hundreds of millions.

For a transaction, its features are encoded into embedding vectors and summed together to generate a transaction embedding. The embeddings of transactions within a sequence are stacked to form a matrix $H_0 \in \mathbb{R}^{N \times d}$, where $N$ is the length of the transaction sequence.

*3.2.2* ***Transformer:*** As shown in Figure 2, the Transformer [31] architecture consists of $L$ transformer layers, which can be formalized as:

$$H_l^{'} = \text{Attention}\left(H_l W_l^Q, H_l W_l^K, H_l W_l^V\right) \quad (1)$$

$$\text{Attention}(Q, K, V) = \text{softmax}\left(\frac{QK^\top}{\sqrt{d}}\right)V \quad (2)$$

$$H_{l+1} = [\text{FFN}(h_l^1); \cdots ; \text{FFN}(h_l^t)] \quad (3)$$

$$\text{FFN}(x) = \text{GELU}(x W_l^1 + b_l^1) W_l^2 + b_l^2 \quad (4)$$

where the projection matrices $W_l^Q, W_l^K, W_l^V, W_l^1, W_l^2 \in \mathbb{R}^{d \times d}$, and bias vectors $b_l^1$ and $b_l^2 \in \mathbb{R}^{d \times 1}$ are trainable parameters for the $l$-th Transformer layer. Here we omit the multi-head mechanism to facilitate description.

The time complexity for $L$-layer Transformer computations is $O(L \cdot N^2 \cdot h \cdot d)$, where $L$, $N$, $h$, and $d$ represent the number of Transformer layers, the length of sequence, the number of heads in self-attention, and the hidden dimension, correspondingly.

## 3.3 Pre-training

There are several well-known tasks to pre-train LMs in NLP, such as next token prediction of GPT [25], masked token prediction of BERT [19], replaced token detection of ELECTRA [5], *etc*. Here we adopt a task named masked address prediction [15] to pre-train a BERT-like LM.

As illustrated in Figure 2, given a transaction sequence, $r\%$ of transactions are randomly selected. The address features of selected transactions are replaced with a special token [MASK], and the sequence is passed through the LM to generate transaction representations. For a transaction whose address is masked, we use

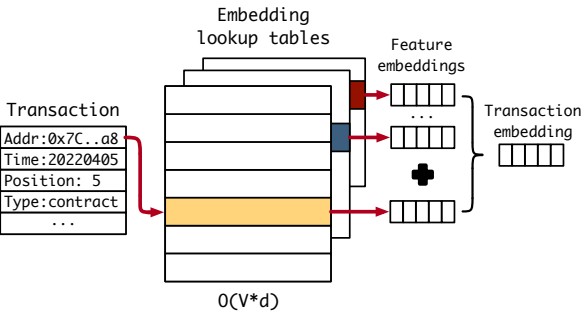

**Figure 3: Encode features into embedding vectors via embedding lookup tables.**

$h_m \in \mathbb{R}^d$ to denote the representation produced by Transformer, which includes its bidirectional context information and is utilized to predict its masked address. Specifically, a contrastive loss is adopted as the objective function:

$$L = -\frac{1}{|\mathbb{M}|} \sum_{m \in \mathbb{M}} \log\left(\frac{\exp(h_m^\top \cdot a_p)}{\exp(h_m^\top \cdot a_p) + \sum_{n \in \mathbb{N}} \exp(h_m^\top \cdot a_n)}\right) \quad (5)$$

where $\mathbb{M}$ is the masked address set in sequences, $a_p$ is its address embedding (positive address), $\mathbb{N}$ is the negative address set and $a_n$ is the address embedding of a different address (negative address). Optimizing Eq. 5 essentially entails encouraging $h_m$ to be close to its address embedding $a_p$, and distant from $a_n$ in the latent space.

## 3.4 Fine-tuning

For an account-level classification task, such as phishing account detection or identity inference, we cascade the pre-trained LM with a MLP classifier, which takes the representation of the self-transaction $h_s$ as input. $h_s$ represents the entire sequence and the EOA. The predicted probability $\hat{y}$ of the EOA being a fraud account is given by:

$$\hat{y} = \text{Sigmoid}\left(\text{MLP}\left(h_s\right)\right) \quad (6)$$

The objective loss is the negative log-likelihood function as:

$$L = -\frac{1}{|\mathcal{D}|} \sum_{(\hat{y},y) \in \mathcal{D}} (y \log \hat{y} + (1 - y) \log(1 - \hat{y})) \quad (7)$$

where $\mathcal{D}$ is the training dataset, and $y \in \{1, 0\}$ is the ground-truth label.

## 4 ZIPZAP

ZIPZAP is a framework that offers parameter and computational efficiency through two strategies: *frequency-aware compression* and *asymmetric training*.

### 4.1 Frequency-aware Compression

*4.1.1* ***Motivation:*** Figure 4 illustrates the parameter proportion of the address embedding lookup table in the entire model. Clearly, the lookup table constitutes 99% of parameters when the number of address approaches million-level. Consequently, compressing the LM essentially entails compressing the address embedding lookup table.

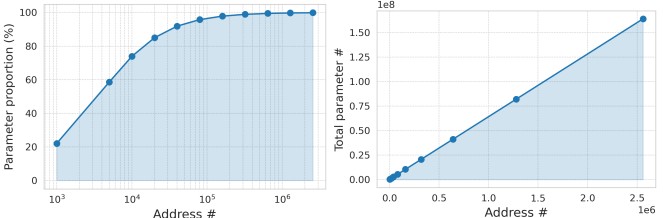

**Figure 4: The parameter proportion of the address embedding lookup table in the entire LM reaches 99% when the address number approaches million-level. The total parameter number scales linearly to the address number.**

Previous studies [15, 20, 40] have demonstrated that the distribution of frequency of address occurred in transactions follows a power-law distribution. As shown in Figure 5(a), a small number of addresses occurs frequently, whereas the majority of addresses occurs only a few times. As described in Section 3.2.1, the parameters of an address embedding can only be *retrieved* and *trained* when the associated address appears at transactions. This indicates that the embedding parameters for Ethereum addresses with low occurrence frequencies, which constitute the *majority* of Ethereum addresses, are trained only a few times in one epoch.

Limited training times result in the majority of address embeddings being located close to their initialization points in the hidden space. In Figures 5(b) we plot the $l_1$ norm of address embeddings after pre-training. It is evident that the $l_1$ norm decreases as the frequency decreases as well, suggesting that the embeddings of low-frequency addresses remain closer to their initial locations. This observation further implies that low-frequency addresses, which make up the majority of the address space, cannot fully exploit the capability of high-dimensional embeddings to represent their knowledge as high-frequency addresses do.

*4.1.2  **Frequency-aware Compression:*** We approach the compression by taking frequency as a signal to assign address embeddings with various dimensions. The compression method consists of two phases: *space partitioning* and *dimension decay*.

**Space Partitioning:** First, we sort the addresses based on frequency in descending order and index them within the range $[0, max)$. Second, we divide the address space into $K$ partitions. The principle for determining the upper and lower bounds of each partition is to keep the sums of address frequencies across different partitions equal, which guarantees that the training times for each partition are equal:

$$F_k = \sum_{j \in P_k} f_j = \frac{1}{K} \cdot \sum_{1}^{K} F_i \tag{8}$$

where $P_k$ is the k-th partition, $F_k$ is the total frequency of addresses in $P_k$, and $f_j$ is the frequency of address $j$ within $P_k$. In this case, given an address, the probability of it being selected from different buckets is all the same. We plot an 4-partition division example in Figure 6(c), where a partition with a larger index covers a much larger range of addresses due to the characteristic of power-law distribution, *i.e.*, the partition range increases exponentially as the partition index increases.

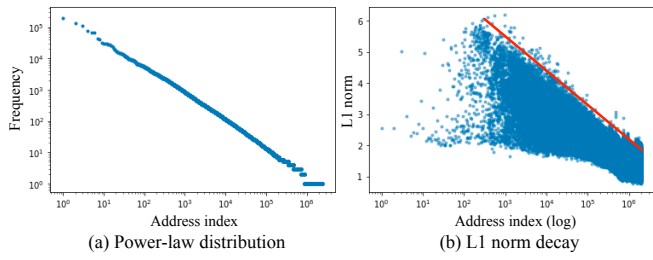

**Figure 5: Addresses are indexed by frequency in descending order. (a) Address frequency follows a power-law distribution. (b) The l1 norm of pre-trained address embeddings decays.**

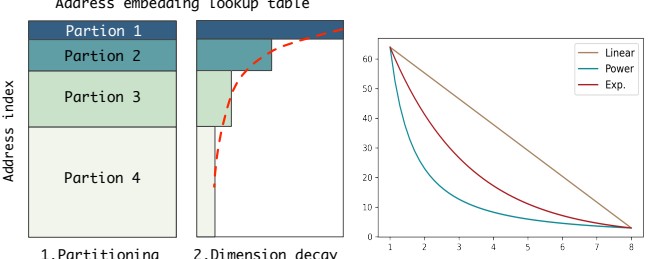

(a) Frequency-aware compression  (b) Dimension decay functions

**Figure 6: Frequency-aware compression.**

**Dimension Decay:** As illustrated in Figure 6(a), our goal is to allocate smaller dimensions to partitions as their indices increase. To determine the optimal relationship between the embedding dimension and the partition index, we propose three dimension decay functions *w.r.t.* the partition index $x$:

- *Linear decay:*

$$d_x = \alpha * (x - 1) + d_u, \alpha = (d_l - d_u)/(K - 1) \tag{9}$$

- *Exponential decay:*

$$d_x = d_u * \alpha^{x-1}, \alpha = (d_l/d_u)^{1/(K-1)} \tag{10}$$

- *Power decay:*

$$d_x = d_u * x^\alpha, \alpha = \log_K(d_l/d_u) \tag{11}$$

where $d_x$ is the dimension of the $x$-th partition $P_x$, $x \in [1, 2, ..., K]$, $d_u$ (upper) is the original (largest) dimension, and the $d_l$ (lower) is the smallest dimension. When $d_u = d_l$ and $B = 1$, it degrades into a uniform embedding dimension. In Figure 6(b) we plot their corresponding figures to demonstrate the varying degrees of decay ($d_u$=64, $d_l$=3). Given the same $d_u$ and $d_l$, we observe that the power decay strategy reaches the lowest compression rate.

For address embedding $a_j$ retrieved from the $i$-the partition $P_x$, we multiply it with a partition-wise matrix $V_x \in \mathbb{R}^{d_x \times d_u}$ to transform its dimension to the original $d_u$:

$$a_j = a_j * V_x \tag{12}$$

Table 1 presents the results of applying three dimension decay strategies to compress the language model with $d_u$=64, $d_l$=3, $K$=8, where the $F_1$ score is evaluated on the downstream phishing account detection task. We note that very low compression rates (less than 6%) are achieved by the linear and exponential decay functions. Among three strategies, the exponential decay function

**Table 1: Performance of frequency-aware compression _w.r.t._ three decay strategies. $F_1$ is evaluated on the downstream phishing account detection task. Time denotes the (pre-training) time cost for every 500 batches.**

| Strategy | $F_1$ | Param.# | Comp. Rate | Time | Speedup |
|---|---|---|---|---|---|
| Original | 0.6552 | 153M | 100.0% | 42.82 | 1.0 |
| Linear | 0.6541 | 14M | 9.15% | 31.65 | 1.353x |
| Exp. | 0.6506 | 9M | 5.88% | 31.58 | 1.356x |
| Power | 0.6476 | 8M | 5.23% | 30.94 | 1.384x |

strikes a good balance between the compression rate and the $F_1$ metric, making it the default setting for ZipZap.

**Effect on training acceleration:** Frequency-aware compression speeds up training, resulting in a 1.356x acceleration, because the computation required for backward gradients is reduced due to a significant decrease in the number of parameters.

## 4.2 Asymmetric Training

_4.2.1_ **_Motivation:_** The training of LMs comprises both pre-training and fine-tuning stages. Pre-training is more time-consuming than fine-tuning, as the different optimization goals of two stages: Pre-training tasks [5, 7, 25] model the co-occurrence relationship among transactions, leveraging the abundant self-supervised signals within sequences. In comparison, fine-tuning tasks, such as phishing account detection, regard the transaction sequence as a whole, drawing on supervised signals external to the sequences.

This inconsistency suggests that adopting training-accelerating strategies for pre-training, while reverting to the standard training paradigm for fine-tuning, might not significantly compromise the overall effectiveness of the LM but yield considerable computational savings.

_4.2.2_ **_Lightweight Pre-training:_** As shown in Figure 7, two tactics are proposed and adopted only at pre-training for acceleration: _transaction dropping_ and _cross-layer sharing_.

**Transaction Dropping:** Dropping transactions to shorten input sequences is intuitive based on the observation that sequence length $N$ has a quadratic relationship to the time complexity of transformer computation, _i.e._, $O(L \cdot N^2 \cdot h \cdot d)$. Nonetheless, randomly dropping transactions leads to performance decrease because less co-occurrence between transactions could be modeled. An guidance to drop transactions is the repetitiveness level of transaction within sequences, which can be measured by the Repetitiveness Score (RS) defined as the proportion of transactions whose addresses are repetitive within the sequence:

$$RS(sequence) = 1 - \frac{\text{\# of unique addresses}}{\text{\# of transactions}} \quad (13)$$

As shown in Table 2, for the original transaction sequences, we observe an average RS of 37.2%, which suggests that 37.2% transactions share the same address within sequences in average, indicating the presence of redundant information that could be filtered out to reduce computation. Specifically, for transactions that have the same address within a sequence, we randomly pick one out of them to keep, and drop out all the other repetitive transactions. This dropping strategy squeezes the RS to 0, shortening the average length from 33.78 to 10.78 and expediting the pre-training to 1.243x faster.

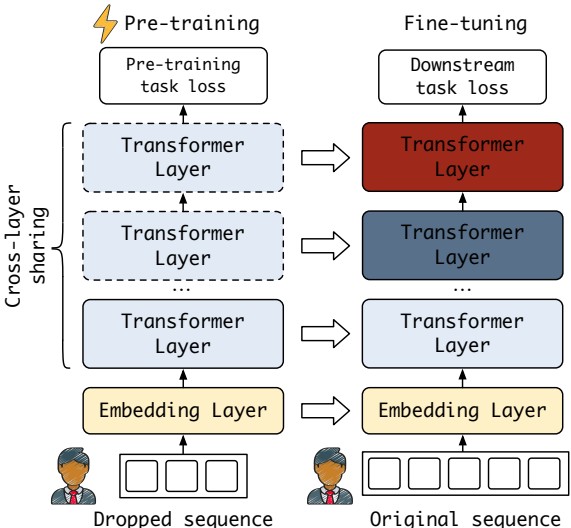

**Figure 7: ZipZap employs transaction dropping and cross-layer sharing during pre-training to enhance computational efficiency, while reverts to the standard training paradigm during fine-tuning to ensure effectiveness.**

**Table 2: Transaction dropping w. asymmetric training.**

| Strategy | RS | Length | $F_1$ | Time | Speedup |
|---|---|---|---|---|---|
| Original | 0.372 | 33.78 | 0.6506 | 31.58 | 1.0 |
| Drop. | 0.0 | 10.78 | 0.6692 | 25.51s | 1.238 |
| Drop.(Asy.) | 0.0 | 10.78 | **0.6742** | **25.51s** | **1.238**x |

**Table 3: Cross-layer sharing w. asymmetric training.**

| Strategy | $F_1$ | Time | Speedup |
|---|---|---|---|
| Original | 0.6742 | 25.51s | 1.0 |
| Cross | 0.6599 | 22.16s | 1.151x |
| Cross(Asy.) | 0.6696 | **22.16s** | **1.151**x |

Furthermore, we observe a 1.86 AP gain of $F_1$ on the downstream task, suggesting that reducing repetitiveness improves the effectiveness of pre-training. This is because the masked address prediction task that models transaction co-occurrence, is susceptible to label leakage caused by high repetitiveness. An alternative solution is to aggregate continuous repetitive transactions into one, which helps reduce repetitiveness, yet cannot handle discontinuous repetitive transactions.

**Cross-layer Sharing:** To further speed up pre-training, we force all the Transformer layers to share parameters across layers during pre-training as shown in Figure 7, _i.e._, trainable parameters in Eq. 1 and Eq. 4 are shared across $L$ Transformer layers. Cross-layer parameter sharing reduces parameters in Transformer, and thus accelerates the back-propagation computation. As demonstrated in Table 3, it brings 1.151x speedup while in the cost of 1.43 AP of $F_1$ drop due to limited model capacity.

_4.2.3_ **_Recovered Fine-tuning:_** Accelerating fine-tuning has minimal impact on the computational cost of the entire training. We recover dropped sequences and lift the cross-sharing constraint to avoid performance decline in fine-tuning.

**Transaction Sequence Recovery:** The optimization goal of fine-tuning is not the same as the pre-training task, suggesting that reducing repetitiveness can not bring improvement for fine-tuning, yet downgrades its performance since those repetitive transactions still carry valuable information for downstream tasks, such as the temporal patterns of user behavior. As a result, we restore the dropped sequences for fine-tuning, resulting in a **0.53** AP lift of $F_1$ for the downstream task as presented in Table 2.

**Unconstrained Transformer Layers:** Adhering to the idea of asymmetric training, we lift the constraint of cross-layer sharing by fine-tuning $L$-Transformer layers independently. From Table 3 we can observe that removing the constraint during fine-tuning brings **0.97** AP lift of $F_1$, a considerable compensation for downgrade caused by cross sharing.

## 5  EXPERIMENT

### 5.1  Experimental Setup

*5.1.1  **Dataset:*** We pre-train the LM on transaction datasets and fine-tune it for detecting phishing scams (accounts), one of the most pervasive frauds on Ethereum. We collected 2,746 phishing accounts (EOAs) from Etherescan that were identified and labeled by users and security companies, serving as positive samples. For negative samples (benign EOAs), we generate three datasets named $\mathcal{D}_S$, $\mathcal{D}_L$, and $\mathcal{D}_{XL}$ by randomly collecting three sets of EOAs and the transactions in which these EOAs were involved from Jan. 2017 to Jan. 2023. Among them, $\mathcal{D}_S$ and $\mathcal{D}_L$ is used for performance comparison, while $\mathcal{D}_{XL}$ is used for large-scale experiments.

The statistics are reported in Table 4, where the "# of EOA" column represents the number of EOAs for which we generate transaction sequences. The "# of transaction" column represents the total number of transactions collected. The "# of address" represents the total number of addresses involved in these transactions, which is equal to the size of the address embedding lookup table. The "Length" column represents the average number of transactions in transaction sequences. The "Neg./Pos." column represents the ratio of benign accounts to phishing accounts.

*5.1.2  **Baselines:*** To measure effectiveness, three types of competitors are compared: 1) Graph learning methods, including Deep-Walk [23], Trans2Vec [36], Diff2Vec [28], Role2Vec [1]; 2) GNN methods, including GCN [18], GAT [32], GraphSAGE [12]; 3) Language models, including BERT (BERT4ETH [15]) and ALBERT [19].

To measure computational efficiency, two types of baselines are involved: 1) Efficient pre-training methods, including ALBERT, Progress Stack [10] and Token Bypass [13]; 2) Efficient Transformers including Longformer [2], Linformer [34] and Performer [4].

To measure parameter efficiency, we compare ZipZap against Learnable Embedding [22] and embedding factorization used by ALBERT. For fairness of comparison, we apply them on the address embedding layer only, with ZipZap as the backbone model.

*5.1.3  **Implementation:*** For LM-based methods including ZipZap, BERT, ALBERT, Progress Stack, Token Bypass, Longformer, Linformer and Performer, the number of Transformer layers is set to 8, the number of heads for self-attention is set to 2 and the maximum sequence length $N$ is set to three times the average length of input sequences. Masked address prediction (Section 3.3) is adopted as

### Table 4: Statistics of datasets

| Dataset | # of EOA | # of trans. | # of address | Length | Neg./pos. |
|---|---|---|---|---|---|
| $\mathcal{D}_S$ | 314,256 | 10,422,570, | 2,128,180 | 33.78 | 114:1 |
| $\mathcal{D}_L$ | 938,176 | 35,894,143 | 6,104,218 | 38.26 | 342:1 |
| $\mathcal{D}_{XL}$ | 3,127,997 | 110,591,442 | 19,004,544 | 35.42 | - |

### Table 5: Effectiveness comparison for fixed training.

| Dataset | | $\mathcal{D}_S$ | | | $\mathcal{D}_L$ | |
|---|---|---|---|---|---|---|
| Method | Pre. | Rec. | $F_1$ | Pre. | Rec. | $F_1$ |
| DeepWalk | 0.2486 | 0.1778 | 0.2074 | 0.1499 | 0.1253 | 0.1365 |
| Trans2Vec | 0.1495 | 0.1391 | 0.1441 | 0.0839 | 0.0824 | 0.0831 |
| Diff2Vec | 0.2556 | 0.1713 | 0.2051 | 0.1566 | 0.1110 | 0.1299 |
| Role2Vec | 0.2770 | 0.2113 | 0.2398 | 0.1890 | 0.1323 | 0.1557 |
| GCN | 0.3152 | 0.2219 | 0.2605 | 0.2077 | 0.1424 | 0.1690 |
| GSAGE | 0.2817 | 0.2404 | 0.2594 | 0.1988 | 0.1554 | 0.1744 |
| GAT | 0.3215 | 0.2519 | 0.2825 | 0.2284 | 0.1663 | 0.1917 |
| BERT | 0.5447 | 0.3632 | 0.4358 | 0.3808 | 0.3140 | 0.3442 |
| ALBERT | 0.5322 | 0.3430 | 0.4171 | 0.3662 | 0.2851 | 0.3206 |
| ZipZap | **0.5694** | **0.3870** | **0.4608** | **0.4239** | **0.3303** | **0.3713** |

### Table 6: Effectiveness comparison for fine-tuning.

| Dataset | | $\mathcal{D}_S$ | | | $\mathcal{D}_L$ | |
|---|---|---|---|---|---|---|
| Method | Pre. | Rec. | $F_1$ | Pre. | Rec. | $F_1$ |
| BERT | 0.7191 | 0.6017 | 0.6552 | 0.6260 | 0.4867 | 0.5476 |
| ALBERT | 0.6823 | 0.5805 | 0.6273 | 0.5750 | 0.4613 | 0.5119 |
| ZipZap | **0.7374** | **0.6132** | **0.6696** | **0.6406** | **0.5011** | **0.5623** |
| **w/o pre-training** | | | | | | |
| BERT | **0.5559** | **0.4482** | **0.4919** | **0.3728** | 0.2940 | **0.3287** |
| ALBERT | 0.5310 | 0.4275 | 0.4737 | 0.3387 | 0.2831 | 0.3084 |
| ZipZap | 0.5355 | 0.4410 | 0.4837 | 0.3508 | **0.3043** | 0.3259 |
| $\text{ZipZap}_D$ | 0.5177 | 0.4325 | 0.4713 | 0.3508 | 0.2872 | 0.3159 |

the pre-training task for all these methods. During pre-training, the masking ratio is set to 80% to prevent label leakage. During fine-tuning, a 2-layer MLP with a hidden dimension of 128 is cascaded as the classifier in Eq. 6. For frequency-aware compression of ZipZap, the number of partition $K$ is set to 8, the maximum dimension $d_u$ is set to 64 and minimum dimension $d_l$ is set to 3. A batch size of 256, a dropout ratio of 20%, and a hidden dimension of 64 are used for all approaches. More details on implementation and hyper-parameter settings can be found in Appendices B to enhance the reproducibility.

**Hardware:** Experiments are conducted on a standard NVIDIA RTX 3090 GPU with 24GB memory.

### 5.2  Effectiveness Comparison

All baselines are self-supervisedly pre-trained on $\mathcal{D}_S$ and $\mathcal{D}_L$, and evaluated for phishing account detection *w.r.t.* two strategies, fixed training and fine-tuning. For fixed training, the pre-trained model is utilized as a feature extractor to generate account representations, followed by individually training a MLP classifier for classification.

Table 7: Comparison with computation-efficient methods. Time is the average time cost (in seconds) for 500 batches.

| Dataset | $\mathcal{D}_S$ | | | | | | $\mathcal{D}_L$ | | | | | |
|---|---|---|---|---|---|---|---|---|---|---|---|---|
| Method | Precision | Recall | $F_1$ | Param.# | Time | Speedup | Precision | Recall | $F_1$ | Param.# | Time | Speedup |
| BERT | 0.7191 | 0.6017 | 0.6552 | 153M | 42.82s | 1.0 | 0.6260 | 0.4867 | 0.5476 | 409.6M | 69.25s | 1.0 |
| ALBERT | 0.6823 | 0.5805 | 0.6273 | 19.3M | 27.40s | 1.56x | 0.5750 | 0.4613 | 0.5119 | 48.3M | 33.45s | 2.07x |
| ProgStack | 0.7130 | 0.5969 | 0.6498 | 153M | 33.88s | 1.26x | 0.6192 | 0.4768 | 0.5387 | 409.6M | 56.76s | 1.22x |
| TokenBypass | 0.7145 | 0.5695 | 0.6338 | 153M | 35.07s | 1.22x | 0.6003 | 0.4756 | 0.5307 | 409.6M | 60.93s | 1.17x |
| Longformer | 0.6820 | 0.5883 | 0.6317 | 153M | 42.78s | 1.00x | 0.5797 | 0.4769 | 0.5233 | 409.6M | 69.74s | 0.99x |
| Linformer | 0.6780 | 0.5847 | 0.6279 | 153M | 42.02s | 1.02x | 0.6085 | 0.4613 | 0.5247 | 409.6M | 66.42s | 1.04x |
| Performer | 0.6602 | 0.5835 | 0.6205 | 153M | 55.96s | 0.77x | 0.5711 | 0.4579 | 0.5083 | 409.6M | 87.19s | 0.79x |
| ZipZap | **0.7374** | **0.6132** | **0.6696** | **9M** | **22.07s** | **1.94x** | **0.6406** | **0.5011** | **0.5623** | **22.8M** | **22.71s** | **3.05x** |

For fine-tuning, the model is trained with a cascaded MLP classifier together. Each experiment is repeated five times and the best $F_1$ score is reported. The threshold is set between 0.2 to 0.4.

Table 5 presents the results of the fixed training strategy. As there is no fine-tuning involved, ZipZap takes dropped sequences as input for fixed-training to maintain the consistency. From the table, the first observation is that LMs outperform graph-based approaches by a large margin, indicating the superior modeling capabilities of the Transformer and the importance of capturing sequential and transaction-level information. The second observation is that ZipZap slightly outperforms its original LM, BERT. The improvement primarily comes from addressing the label leakage problem via transaction dropping as described in Section 4.2.2. The performance difference between two datasets is caused by the varying negative-to-positive sample ratios.

Table 6 presents the results after fine-tuning, where we omit the graph-based methods since they underperform. The first three rows show the results of fine-tuning with pre-training, which demonstrate that pre-training can bring huge improvements over competitors compared to results in Table 5. Additionally, ZipZap still outperforms BERT model with **1.44** and **1.47** AP on both datasets, yet the performance gap is decreased compared to fixed-training, suggesting that fine-tuning narrows the performance gap caused by pre-training.

To investigate the benefits of pre-training and fine-tuning separately, we ablate the pre-training process and presents the results of directly trained on the phishing detection task in the last five rows of Table 5, where $\text{ZipZap}_D$ is trained on dropped sequences and ZipZap is trained on recovered sequences. The results show that the ZipZap performs worse than BERT due to the frequency-aware embedding compression. Moreover, we observe that transaction dropping decreases the performance by comparing ZipZap with $\text{ZipZap}_D$, suggesting that the same strategy poses an *opposite* effect for the pre-training and fine-tuning stages, which further justifies the idea of *asymmetric training*.

## 5.3 Efficiency Comparison

*5.3.1 Computational Efficiency Comparison:* ALBERT, ProgStack and TokenBypass speed up training from three aspects: reducing the number of parameters, progressively copying pre-trained parameters for initialization, and reducing computation for trivial tokens. Another type of methods is adopting efficient Transformer

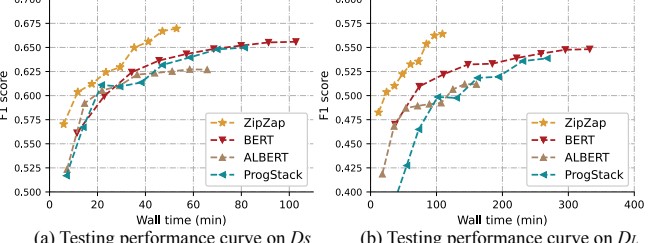

(a) Testing performance curve on $D_S$     (b) Testing performance curve on $D_L$

Figure 8: Testing $F_1$ on the phishing detection task *w.r.t.* the pre-training time (checkpoints).

as the backbone, which can accelerate the self-attention calculation. In this experiment we only compare the speed of pre-training, as the time cost of fine-tuning is negligible contrasted with pre-training.

Table 7 compares the efficiency and effectiveness, where Time is the average (pre-training) time cost for 500 batches, $F_1$ is the result after fine-tuning. Our observations are as follows: (1) Efficient Transformers are less efficient and effective compared to the methods (ALBERT, ProgStack, and TokenBypass) designed for accelerating pre-training. This is because the average sequence length in our scenario is not very long, making the acceleration of self-attention insignificant, and additional operations involved even lead to negative effects. (2) While ALBERT, ProgStack, and TokenBypass improve efficiency, they also result in a decrease in the $F_1$ score. (3) ZipZap offers both efficiency and effectiveness, as it provides a $F_1$ gain and 1.94x and 3.05x speedup on the two datasets. The improvement in computational efficiency comes from two factors: (i) Reduction of backward gradient computation because 94% of parameters are reduced by frequency-aware compression. (i) Reduction of Transformer computation because of transaction dropping and cross-layer sharing.

Furthermore, Figure 8 plots the $F_1$ scores of ZipZap, BERT, ALBERT, and ProgStack *w.r.t.* pre-training time. For each pre-training checkpoint, we fine-tune it on the downstream task to evaluate its $F_1$ performance. It can be observed that ZipZap reaches a higher $F_1$ score in a shorter pre-training time, and its advantage over the other competitors becomes more pronounced as the dataset size increases. The reason is because the address embedding lookup table for $\mathcal{D}_L$ is 2.7x larger than $\mathcal{D}_S$, resulting a better benefit from reducing the backward gradient computation.

Table 8: Comparison with parameter-efficient methods. (-) denotes out-of-memory (OOM).

| Dataset | $\mathcal{D}_S$ | | | | | | $\mathcal{D}_L$ | | | | | |
|---|---|---|---|---|---|---|---|---|---|---|---|---|
| Method | Precision | Recall | $F_1$ | Sparsity | Param.# | Comp. Rate | Precision | Recall | $F_1$ | Sparsity | Param.# | Comp. Rate |
| BERT | 0.7191 | 0.6017 | 0.6552 | 0% | 153M | 100.0% | 0.6260 | 0.4867 | 0.5476 | 0% | 409.6M | 100.0% |
| ALBERT | 0.6823 | 0.5805 | 0.6273 | 0% | 19.3M | 12.61% | 0.5750 | 0.4613 | 0.5119 | 0% | 48.3M | 11.79% |
| Factorization | 0.7052 | 0.5822 | 0.6378 | 0% | 19.3M | 12.61% | 0.6044 | 0.4811 | 0.5357 | 0% | 48.3M | 11.79% |
| LearnEmbed | 0.6662 | 0.5447 | 0.5993 | 63.40% | 307.2M | 200.8% | - | - | - | - | ∼ 820M | - |
| LearnEmbed* | 0.6398 | 0.5043 | 0.5640 | **87.27%** | 307.2M | 200.8% | - | - | - | - | ∼ 820M | - |
| ZIPZAP | **0.7374** | **0.6132** | **0.6696** | 0% | **9M** | **5.88%** | **0.6406** | **0.5011** | **0.5623** | 0% | **22.8M** | **5.57%** |

Table 9: Statistics of large-scale datasets derived from $\mathcal{D}_{XL}$ with different filtering rules.

| Dataset | # of address | Length | RS |
|---|---|---|---|
| $\mathcal{D}_{XL}$ | 19,004,544 | 35.42 | 0.372 |
| $\mathcal{D}_{XL1}$ | 13,626,560 | 78.01 | 0.565 |
| $\mathcal{D}_{XL2}$ | 11,649,431 | 143.35 | 0.652 |

*5.3.2 Parameter Efficiency Comparison:* Factorization and LearnEmbed are two representative approaches for embedding compression. For fairness in comparison, we apply them to ZIPZAP by replacing our frequency-aware compression with their compression techniques, and all other conditions remain the same. LearnEmbed and LearnEmbed* are initialized with different masking threshold values (-5 and -4) that lead to varying levels of sparsity. Sparsity is defined as the percentage of non-zero parameters in the address embedding layer.

Table 8 presents the results of parameter efficiency comparison. It can be observed that ZIPZAP outperforms existing methods, yielding a significant improvement with a 3.18 AP increase over Factorization and a 6.73 AP lower compression rate on $\mathcal{D}_S$. This enhancement is solely attributed to the frequency-aware embedding compression, highlighting the importance of considering address occurrence frequency in embedding dimension. ALBERT also shows good parameter efficiency through its adoption of the factorization technique. On the other hand, the performance of LearnEmbed is not satisfactory. Although 87% of its parameters are pruned to zero, the unstructured pruning is unfriendly to hardware and cannot truly reduce memory usage. Moreover, the learnable thresholding introduces extra parameters and computation overhead, resulting in slower pre-training and requiring twice the parameters of BERT. For this reason, it causes an out-of-memory (OOM) error on the experimental hardware for the experiments on $\mathcal{D}_L$.

## 5.4 Ablation Study

We investigate the impact of two strategies proposed for ZIPZAP, *i.e.,* frequency-aware compression and asymmetric training in terms of computational efficiency on larger-scale datasets. To this end, we create another two datasets, $\mathcal{D}_{XL1}$, and $\mathcal{D}_{XL2}$, by filtering out EOAs with fewer than 10, and 20 transactions from $\mathcal{D}_{XL}$, respectively. The filtering rules lead to two datasets with different statistics as presented in Table 9, where $\mathcal{D}_{XL}$ has the highest number of addresses, yet the shortest sequence length and $\mathcal{D}_{XL}$ has the lowest number of addresses, yet the longest sequence length. A desirable characteristic of Ethereum transaction is that as the sequence length

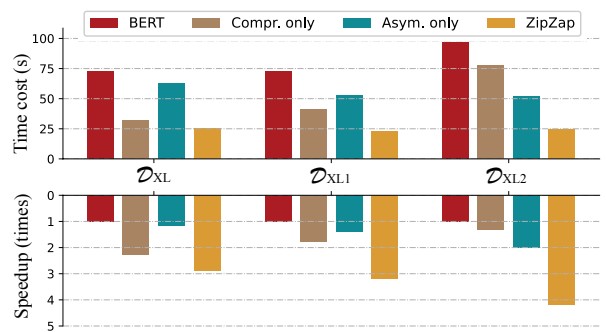

Figure 9: Ablation study on large-scale datasets. Embedding compression contributes more than asymmetric training when the sequence is shorter and number of address is larger.

increases, the Repetitiveness Score becomes higher, making ZIPZAP more advantageous from transaction dropping. We evaluate four models, *i.e.,* the base BERT, ZIPZAP w/ compression only, ZIPZAP w/ asymmetric training only and ZIPZAP. To prevent the base model encountering the OOM error, the batch size and hidden dimension are set to half of the original hyper-parameters.

In Figure 9, we report the time cost for 500 batches and speedup for pre-training. By comparing ZIPZAP w/ compress. with BERT, we observe that the improvement in computation-efficiency brought by compression is the most significant for $\mathcal{D}_{XL}$, which owns the largest address embedding layer. As the number of addresses decreases, the speedup also decreases. Additionally, we observe that the contribution of asymmetric training increases as the sequence length increases ($\mathcal{D}_{XL}$->$\mathcal{D}_{XL1}$->$\mathcal{D}_{XL2}$), and surpasses ZIPZAP w/ compress. on $\mathcal{D}_{XL2}$. Notably, under different settings, the time cost of ZIPZAP remains relatively stable, which is a desired feature for large-scale applications.

## 6 CONCLUSION

In this study, we present ZIPZAP, an innovative framework crafted for efficient and effective training of LMs tailored for Ethereum data. Equipping with the frequency-aware compression technique, ZIPZAP enjoys a remarkable 94% reduction in parameters from the original LM by leveraging frequency as the signal for dimension allocation. With the asymmetric training approach, ZIPZAP optimizes both the speed of pre-training and the efficacy of fine-tuning. Coupled with two strategies, ZIPZAP expedites the entire training process up to **3** times faster on large-scale real world datasets. Overall, our study sheds light on bridging state-of-the-art representation learning techniques with large-scale fraud detection applications.

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

# A REPRODUCIBILITY

To improve reproducibility, we include the key hyperparameters used in ZipZap in Table 10.

**Table 10: Hyper-parameters of ZipZap**

| Phase | Hyper-parameter | Value |
|---|---|---|
| Basic | Transformer layer | 8 |
| | Head number | 2 |
| | Hidden size ($d_u$) | 64 |
| | $d_l$ | 3 |
| | Bucket number | 8 |
| | Decay strategy | Exp. |
| Pre-training | Learning rate | 1e-4 |
| | Masking ratio | 80% |
| | Dropout ratio | 20% |
| | Data duplicate times | 10 |
| | Epoch | 5 |
| | Batch size (seqs) | 256 |
| Fine-tuning | MLP hidden size | 128 |
| | Learning rate | 3e-4 |
| | Dropout ratio | 20% |
| | Data duplicate times | 1 |
| | Epoch | 1 |
| | Batch size (seqs) | 256 |

# B IMPLEMENTATION DETAIL

For graph-based methods, we adopt the self-supervised task proposed in DeepWalk [23]. We set the number of walks per node to 10, the walk length to 20, and the context window size to 5. The number of GNN layers is set to 2, with a neighbor sample size of 50. For all methods, the batch size is set to 256, the dropout rate to 20%, and the hidden dimension to 64, based on empirical hyperparameter tuning. DeepWalk-based methods are implemented using Genism [27], and GNN-based methods are implemented using DGL [33].

The hyper-parameter settings for LM-based methods are kept consistent with ZipZap as outlined in Table 10, and the maximum sequence length is set to three times the average sequence length, as sequence length follows a power-law distribution. Specifically, for ALBERT, the factorization size for the embedding is set to 8. For Linformer, the factorization size for the self-attention mechanism is set to 16. For Performer, the number of multi-head self-attention is set to 8 to achieve accurate attention estimation. The original TokenBypass selects 50% of tokens except special tokens to bypass. For TokenBypass we follow its original setting by only masking 15% of the tokens, and select the left 50% of tokens to bypass.

For parameter-efficient methods including LearnEmbed and embedding factorization, we adopt ZipZap as the backbone. In the case of Embedding Factorization, the factorization size is set to 8, which is consistent with ALBERT. For LearnEmbed, we set the initial masking threshold to -5 and -4, referred to as LearnEmbed and LearnEmbed* respectively.

All hyper-parameters are carefully tuned to ensure the best performance.

