# OpenReview forum: "ZipZap: Efficient Training of Language Models for Ethereum Fraud Detection"
_ACM.org/TheWebConf/2024/Conference — TheWebConf24 Oral_

### Official Review · Reviewer_5rb9 · 2023-11-21

**Novelty:** 5
**Technical Quality:** 4

**Review:**

**Quality.** This paper designs a novel efficient training method for training language models (LMs) on Ethereum account data for fraud detection. The key idea is to adopt a frequency-aware compression technique in embedding layer to reduce the model parameters. The paper is well-motivated and the technique applied is sound.

**Clarify.** The paper is well written and easy to follow.

**Significance.** The experiments show that the proposed framework ZipZap achieves better efficiency-effectiveness trade-off than previous methods.

The paper has the following pros and cons:

Pros:
1. The paper is well motivated and proposes an effective training framework for training LMs on Ethereum data.
2. The paper is well written and easy to follow.

Cons:
1. Lack of discussion of extending such compression method to other domains/tasks. The generalizability is not clear.
2. More recent Transformer-based LMs can be tested such as T5.

**Questions:**

Can you discuss what other applications of this framework? And why only tested BERT-based LMs on Ethereum data?

**Reviewer Confidence:**

2: The reviewer is willing to defend the evaluation, but it is likely that the reviewer did not understand parts of the paper

**Scope:**

3: The work is somewhat relevant to the Web and to the track, and is of narrow interest to a sub-community

---

### Official Review · Reviewer_NckV · 2023-11-22

**Novelty:** 5
**Technical Quality:** 5

**Review:**

Summary:

The paper presents novel techniques to decrease the size of embedding lookup features and reductions in the training time required to produce said embeddings. The authors propose the use of frequency-aware compression to optimize the dimensionality of address embeddings during pre-training, while transaction dropping is investigated to reduce the size of sequences during fine-tuning.


Comments:

*Motivating efficiency optimization
While multiple techniques are presented to decrease the time taken to pre-train the address embeddings, the authors do not provide a concrete motivation for the benefits of having faster pre-training.

Further, the argument at the end of the introduction isn't that convincing as a dataset of 32 billion parameters (assuming 32-bit float) would only take ~256GB of storage.

I would suggest the authors revisit their motivation for improving training and parameter efficiency beyond assisting researchers. Typically, there is a longstanding focus on improving inference efficiency as an organization can provide as many resources as they need to train the model once, then save on inference costs in perpetuity.


*Frequency-aware compression

It's not clear to the reader how the partition-wise matrix V_x is defined. There are missing details as to why the number of partitions K is set to 8. Please define V_x and intuition/result as to why K=8 was the most suitable option.


*On the use of BERT to encode addresses

Part of the problem presented in the BERT approach to encoding entire addresses is the size of the address space. However this approach overlooks a benefit provided by BERT-based models which can decompose english words into sub-word n-grams of consecutive letters. The n-grams permit the optimization of commonly occurring words, while still handling unseen vocabulary (in this setting, potentially unseen addresses), while reducing the number of encoded elements. In this address setting the same decomposition may be applies, for example, a 2-gram hex character pair would only have 256 encodings, and 4096 for a 3-gram model, etc. . This is unclear how this approach sits with respect to training efficiency, however the number of unique parameters in the lookup table are greatly reduced.

I would suggest the authors add this approach as a comparative measure, or speculate why this approach is not suitable.



*Minor:

- Why is the technique called ZipZap? I could infer a potential reason from BERT4ETH which uses Zipfan, but this is not explicitly noted for readers.?

- Table 1, 2 are missing units for time.

- Sec 5.3.2/ Table 8, What is running out of memory? GPU? RAM?

- Grammar

-- P.g.2 Line 118 "it accounts the majority time of training, while revert to" -> "it accounts for the majority of training time, while reverting to"

-- P.g.2 Line 208 "provide some backgrounds" > "provide some background"

-- P.g.7 Line 754 "Another type of methods is" -> "Another type of method is"

**Questions:**

1. Why K=8 was the most suitable option for frequency compression?
2. Are there any other motivating factors for reducing training time?
3. Are there reasons why the address could not be decomposed into sub-word components?

**Reviewer Confidence:**

3: The reviewer is confident but not certain that the evaluation is correct

**Scope:**

3: The work is somewhat relevant to the Web and to the track, and is of narrow interest to a sub-community

---

### Official Review · Reviewer_q42a · 2023-11-22

**Novelty:** 5
**Technical Quality:** 5

**Review:**

The paper presents "ZipZap," a framework designed to enhance the efficiency of training language models (LMs) specifically for detecting Ethereum fraud. The primary challenges addressed in the paper are the high computational and memory costs associated with training LMs on large-scale Ethereum data, which include billions of transactions and millions of addresses.

Strength:

- Trendy topic
- Innovative approach

Weakness:

- Lack some details

**Questions:**

First, for the transaction dropping method, I think the authors should provide more details about comparing different drop rates. For different subjects of training LM, there are also some masks in the dataset, so choosing the mask rate is important. Thus, I suggest the authors should do some ablation study on that.

The techniques and findings are specifically tailored for Ethereum fraud detection. It may not be clear how well ZipZap's methods and results generalize to other domains or types of language model applications outside of blockchain data analysis. I suggest the authors should talk more about the transferability of the proposed method.

**Reviewer Confidence:**

4: The reviewer is certain that the evaluation is correct and very familiar with the relevant literature

**Scope:**

4: The work is relevant to the Web and to the track, and is of broad interest to the community

---

### Official Review · Reviewer_uKS4 · 2023-11-24

**Novelty:** 5
**Technical Quality:** 5

**Review:**

**Summary:**
Language models have been increasingly utilized for fraud detection within Ethereum networks.
This paper proposes an efficient training framework for the language model in fraud detection.
The authors begin by identifying the frequency characteristics of the training data and introduce a novel frequency-aware compression technique.
Subsequently, they considered dropping certain training transactions and employing cross-layer parameter sharing to speed up the training process.
The evaluation demonstrates that this approach surpasses existing baselines, particularly in terms of efficiency.

**Strengths:**
- The paper is well structured and offers a clear workflow.

- The evaluation shows that the proposed method outperforms many existing studies, with a notable improvement in efficiency.

**Weaknesses:**
The design is heavily based on the specific characteristics of the training and is evaluated against a singular type of attack. This raises questions about its generalizability to other types of attack or its effectiveness against for example zero-day attacks.

**Questions:**

- Motivation is the primary concern.
Despite the use of advanced LMs, the recall and precision rates remain modest.
This leads to questioning the motivation behind prioritizing speed enhancements over improving the efficacy of existing learning-based or other traditional approaches.

- Given that the optimization relies on specific characteristics, is the proposed design limited to addressing only phishing scams?

- Whether the method's focus on address frequency as their main factor in Ethereum fraud detection suggests that other elements, such as transaction amounts, are less useful in the detection? Is address frequency the most significant indicator of fraud?

**Reviewer Confidence:**

1: The reviewer's evaluation is an educated guess

**Scope:**

3: The work is somewhat relevant to the Web and to the track, and is of narrow interest to a sub-community

---

### Decision · Program_Chairs · 2024-01-22

**Decision:**

Accept (Oral)

**Comment:**

Summary

 This paper focuses on improving the efficiency of training Language Models for Ethereum fraud detection. The paper proposes a novel training framework that achieves both parameter and computational efficiency during pre-training and fine-tuning. The authors first use frequency-aware compression to optimize the dimensionality of address embeddings. Then, they use transaction dropping and cross-layer parameter-sharing.

 Strengths:

 + Novel Techniques on a Trendy Topic. Using language models for Ethereum fraud detection is increasingly popular. This paper introduces novel techniques to reduce the embedding size and training time to speed up language model training.
 + Significant Performance. ZipZap shows improved parameter and computational efficiency compared to existing methods.
 + Writing and Structure. The paper is well-written and easy to follow.

 Weaknesses:

 - The authors can include more ablation studies of their design components, as well as more clarification/evaluation on extending the proposed method to other domains.